# The Freeze-Thaw Strength Evolution of Fiber-Reinforced Cement Mortar Based on NMR and Fractal Theory: Considering Porosity and Pore Distribution

**DOI:** 10.3390/ma15207316

**Published:** 2022-10-19

**Authors:** Chaoyang Zhang, Taoying Liu, Chong Jiang, Zhao Chen, Keping Zhou, Lujie Chen

**Affiliations:** 1School of Resources & Safety Engineering, Central South University, Changsha 410083, China; 2Hunan Survey and Design Institute of Communication, Changsha 410003, China

**Keywords:** cement mortar, fiber, NMR, fractal theory, prediction model, freeze-thaw

## Abstract

Predicting the strength evolution of fiber-reinforced cement mortar under freeze-thaw cycles plays an important role in engineering stability evaluation. In this study, the microscopic pore distribution characteristics of fiber-reinforced cement mortar were obtained by using nuclear magnetic resonance (NMR) technology. The change trend of *T*_2_ spectrum curve and porosity cumulative distribution curve showed that the freeze-thaw resistance of cement mortar increased first and then decreased with the fiber content. The optimal fiber content was approximately 0.5%. By conducting mechanical experiments, it is found that the uniaxial compressive strength (UCS) of the samples exhibited the ‘upward convex’ evolution trends with freeze-thaw cycles due to cement hydration, and based on fractal theory, the negative correlation between UCS and *D*_min_ was established. Eventually, a freeze-thaw strength prediction model considering both porosity and pore distribution was proposed, which could accurately predict the strength deterioration law of cement-based materials under freeze-thaw conditions.

## 1. Introduction

Cement-based materials have become the dominant raw material for various civil engineering projects because of their superior performance, convenient material acquisition, and low cost [1,2]. However, in high-altitude and high-latitude regions, the complex environment under freeze-thaw conditions has caused many projects to suffer safety hazards after a period of use, and even early decommission. Thus, exploring concrete with great frost resistance becomes the key to solve this problem.

Because of the strong toughness and high elongation, fiber is regarded as an important material for concrete improvement. Zhou et al. [3] compared the mechanical properties of ordinary concrete and fiber-reinforced concrete through model tests, and found that fiber could significantly improve the compressive, flexural, and impact resistance of concrete. Gupta et al. [4] conducted durability evaluation and performance research on fiber-reinforced concrete, and concluded that fiber could improve the wear resistance of concrete, reduce shrinkage rate and improve concrete fluidity. Luo et al. [5] studied the freeze-thaw resistance of fiber-reinforced concrete and found that fiber could improve the strength of concrete, but the strengthening effect gradually failed under cyclic freeze-thaw conditions. Li et al. [6] found that fiber could simultaneously improve the freeze-thaw resistance and the chloride ion penetration resistance of concrete. Although these scholars have studied the performance of fiber-reinforced concrete under freeze-thaw conditions, most of them pay attention to the change of macroscopic performance, and the understanding of its microscopic mechanism is still insufficient.

Since freeze-thaw controls the phase change process of pore water by adjusting the temperature, some scholars have tried to reveal the freeze-thaw damage mechanism from a meso-level perspective [7,8,9,10,11]. For example, Liu et al. [12] used CT scanning technology to detect the damage characteristics of rock meso-structure, and achieved a quantitative description of freeze-thaw damage. Grubesa et al. [13] used three methods, mercury intrusion porosity analysis, X-ray microcomputer tomography analysis, and scanning electron microscope analysis, to analyze the meso-structure changes of mortar samples under freeze-thaw conditions. The results showed that pore connectivity was the most important factor affecting freeze-thaw resistance. Cao et al. [14] analyzed the meso-structure of concrete through Matlab binarization image processing technology and realized the quantitative description of the pore distribution state. However, these methods cannot realize the tracking detection of the same sample during freeze-thaw because the detection process will cause damage to the sample. In recent years, NMR non-destructive technology has been used to detect pore structure, which can obtain pore information by detecting the signal released by water molecules in saturated media [15,16,17]. This technology is becoming a helpful method to study the evolution law of fiber-reinforced concrete under freeze-thaw cycles from a microscopic perspective.

UCS loss under freeze-thaw is an important index to evaluate frost resistance. In recent years, scholars have conducted a lot of research on the deterioration law of material strength under freeze-thaw cycles. For example, based on Lemaitre hypothesis, Wang et al. [18] proposed the deterioration model of concrete under the coupling effect of freeze-thaw and stress by analyzing the change of elastic modulus. Liu et al. [19] proposed a UCS prediction model under freeze-thaw cycles by analyzing the change of P-wave velocity based on elastic-plastic theory and fatigue damage mechanics. Bayram [20] predicted the percentage loss values in uniaxial compression strengths from intact rock properties. These models describe the deterioration process of mechanical properties through the change of macroscopic parameters, but for multiphase medium materials, the frost resistance is often determined by the microstructure. Huang et al. [21] suggested that in addition to elastic modulus and tensile strength, initial porosity was also an important parameter to describe the loss of freeze-thaw strength. Amirkiyaei et al. [22] and Zhang et al. [23] proposed damage models by using the change of porosity before and after freeze-thaw cycles to quantitatively describe the deterioration law of macroscopic mechanical properties. However, Li et al. [24] believed that the porosity could only reflect the total number of pores, and proved through experiments that even if the porosity of the two samples was similar, the uniaxial compressive strength could be nearly doubled. For this reason, some scholars also tried to predict the law of strength deterioration by analyzing the change of pore distribution state [25,26]. Xue et al. [27] found that the change of pore position would lead to the change of sample strength. Huang et al. [28] obtained the relationship between strength loss and pore structure change through MIP test. Deng et al. [29] used the macropore ratio and macropore dimension as variables to establish a strength prediction model. In fact, the sample strength is not only related to the number of pores but also related to the size and distribution of pores. By simply considering the change of porosity or the change of pore distribution state, the model cannot reflect the complete microstructure information, so the current methods have shortcomings in predicting the change of macroscopic strength. The objective of this study is to obtain the microscopic pore structure information of cement mortar through NMR non-destructive technology and to establish a damage degradation model that considers both porosity and pore distribution by analyzing the relationship between the microstructure parameters and the macroscopic mechanical properties, so as to improve the accuracy of freeze-thaw strength degradation model

In this study, cement mortar is used as the research object, and polyester fiber is used as the modified material. Section 2 presents the experimental materials and procedures. Section 3 combines NMR technology to analyze the microscopic pore change process under freeze-thaw cycles, and the evolution process of uniaxial compressive strength is also analyzed. In Section 4, the shortcomings of existing deterioration models are analyzed, and then based on fractal theory, the necessity of considering porosity and pore distribution state is illustrated. In Section 5, a freeze-thaw strength degradation model considering both porosity and pore distribution is proposed and verified.

## 2. Experimental Materials and Procedures

The detailed testing procedures and instruments are illustrated in Figure 1, the experimental instruments used in this research mainly include vacuum saturation system (produced by Niumag Analytical Instrument Co., Ltd. in Suzhou, China), freeze-thaw test system (produced by Donghua Examination Appartus Co., Ltd. in Suzhou, China), NMR analysis system (produced by Niumag Analytical Instrument Co., Ltd. in Suzhou, China) and universal testing machine (produced by Hualong Test Instrument Co., Ltd., in Shanghai, China).

### 2.1. Material Selection and Sample Preparation

Fiber-reinforced cement mortar was made by mixing water, sand, cement, water reducing agent, and polyester fiber in a certain proportion. The single control variable method was used to study the influence of polyester fiber content on the performance of cement mortar. The specific content of each substance is shown in Table 1.

The cement selected for the sample was 42.5 R ordinary Portland cement. The initial setting and final setting times are 70 min and 360 min, respectively. The compressive strength for 28 days is 38.5 MPa, the flexural strength is 7.8 MPa, and the fineness is 3.6%. The sand was local river sand, with an apparent density of 2640 kg/m^3^, bulk density of 1430 kg/m^3^, mud content of 0.8%, and fineness modulus of 2.91. The sand passed through 4.75 mm sieve, and its particle distribution is shown in Figure 2. Water reducing agent was polycarboxylate superplasticizer, water reduction rate of 35%, gas content of 3.4%. Polyester fiber is bundle monofilament, length approximately 10 mm, diameter 13–21 mm, specific gravity 1180 kg/m^3^, tensile strength ≥900 MPa, fracture elongation 10–20%, elastic modulus ≥17,000 MPa.

Restricted by the signal detection coil size of the NMR analysis system, the mortar sample prepared in the experiment is a cube with a side length of 70.7 mm. Cement mortar samples were poured according to the Chinese national standard “Test Method for Long-term Performance and Durability of Ordinary Concrete” (GB/T50082-2009) [30], and cured in the mold for 24 h. Then, the samples were demolded and put into a curing box with a constant temperature of 20 ± 2 °C and a relative humidity of 95%, curing for 28 d.

### 2.2. Freeze-Thaw Treatment

The freeze-thaw cycle test was carried out in accordance with the “slow freezing method” in the third chapter of the standard GB/T50082-2009. The instrument used in the test was TDS-300 freeze-thaw test system, which could automatically control the freeze-thaw cycle through the program controller. The upper limit of the cycle is 999 times, and the allowable temperature range is −40 °C~+ 20 °C. In this experiment, the freezing temperature was set at −20 °C and the melting temperature was set at 20 °C. The freezing/thawing duration was 4 h in each cycle. The relationship between the temperature and time of the test chamber in a single cycle is shown in Equation (1):(1)$$T=\left\{\begin{array}{l} 20-80t+53.33t^{2}-11.85t^{3}\;\;\;\;\;\;\;,0\leq t<1.5\\ -20\;\;\;\;\;\;\;\;\;\;\;\;,1.5\leq t<5.5\\ -20+80\lambda -53.33\lambda^{2}+11.85\lambda^{3}\;,5.5\leq t<7\\ 20\;\;\;\;\;\;\;\;\;\;\;,t\geq 7 \end{array}\right.$$
where λ=t−5.5, T is the temperature of the freeze-thaw test chamber, °C, and t is the time, h.

The samples were soaked in water for 48 h to be completely saturated before freeze-thaw test and then subjected to 25, 50, 75, and 100 freeze-thaw cycles according to the test purpose.

### 2.3. NMR Analysis

Given the principle of NMR technology, the pore information can be detected by the coil only if the pores of the sample are filled with water. Therefore, before the NMR analysis, the samples were saturated with water by the vacuum saturated water system. After 4 h of air pumping, the samples were immersed in water for 24 h. Porosity and *T*_2_ spectrum distribution of saturated samples were obtained by using AniMR-150 NMR analysis system. The magnetic field intensity of the system is 0.3 ± 0.05 T, and the RF pulse frequency range is 2~49.9 MHz.

### 2.4. Uniaxial Compression Tests

Uniaxial compression tests were conducted on WHY-200 microcomputer-controlled automatic pressure testing machine, with the maximum pressure of 200 KN and the accuracy of level 1. The loading rate was set at 0.03 mm/min, and three samples were tested for each experimental condition.

## 3. Experimental Results and Analysis

### 3.1. NMR Microscopic Pore Distribution Characteristics

#### 3.1.1. *T*_2_ Spectrum and Porosity Cumulative Distribution Curve

The principle of NMR porosity measurement is to fill the pores with water and then generate *T*_2_ spectrum by collecting the magnetic resonance signals released by hydrogen nuclei in the water medium. Finally, the porosity can be obtained by appropriate calibration of the *T*_2_ spectrum [31]. Figure 3a shows the *T*_2_ spectrum curve obtained in the experiment. The abscissa *T*_2_ is the transverse relaxation time, which is directly related to the pore size [32], namely:(2)$$T_{2}=\frac{r}{\eta \cdot \rho }$$
where r is the pore diameter, η is the surface relaxation strength, and ρ is the pore shape factor. η and ρ can be regarded as constants, so the larger the *T*_2_ value, the larger the pore size. The ordinate is the porosity corresponding to each relaxation time point *T*_2_, and the larger the value, the larger the porosity occupied by the corresponding pore size. As can be seen from the figure, the *T*_2_ spectrum curve can be roughly divided into two parts. The area with *T*_2_ < 10 is classified as micropores, and the area with *T*_2_ ≥ 10 is classified as macropores. The area enclosed by the *T*_2_ spectrum curve and the coordinate axis is proportional to the total amount of hydrogen atoms, that is, the integral sum of the curve can represent the total amount of fluid in porous media. Therefore, the porosity cumulative distribution curve shown in Figure 3b can be obtained by integral transformation of the curve in Figure 3a.

#### 3.1.2. Changes of NMR Curves under Freeze-Thaw Cycles

Figure 4 shows the changes of NMR characteristic curves with freeze-thaw cycles. By observing the change of cumulative porosity, it can be found that although the fiber content is different, the porosity of four groups shows an increasing trend with freeze-thaw cycles. This is because the foundation porosity of cement mortar is very large which leads to the volume expansion (caused by water-ice phase transformation under saturated state) is greater than the volume shrinkage (caused by low temperature and other factors), so freeze-thaw will lead to an increase in the porosity. The longitudinal distribution of the porosity cumulative distribution curves shows that the distance between the curves increases with the increase of cycles, indicating that the deterioration damage of freeze-thaw on the pore structure is gradually enhanced. At the same time, there are differences in the porosity cumulative distribution curves with different fiber contents, mainly manifested as the influence of fiber content on the porosity change rate. Porosity change rate κ is defined as:
(3)$$\kappa =\frac{P(N)-P_{0}}{P_{0}}\times 100\%$$
where P(N) is the porosity after N freeze-thaw cycles, and $P_{0}$ is the porosity at 0 freeze-thaw cycle.

Table 2 lists the porosity change rates of four groups after different cycles. It is found by data comparison that the porosity change rates of four groups after 100 cycles have the following relationship:(4)$$A_{1}=2.05A_{2}=3.43A_{3}=1.04A_{4}$$

That is, the freeze-thaw resistance of samples is A3 > A2 > A4 > A1 from strong to weak, indicating that the addition of fiber can effectively improve the freeze-thaw resistance of cement mortar, while when the fiber content exceeds 0.5%, excessive mixing amount will hinder the performance of fiber. In fact, this phenomenon can be explained by Figure 5. Figure 5 shows the *T*_2_ spectral distribution curves of cement mortars with different fiber contents after 100 freeze-thaw cycles. As the fiber content increases from 0 to 0.5% (A1–A3), the curves corresponding to macropores and micropores both drop, indicating that the fibers dispersed in the cement mortar can effectively fill the gaps between particles. When the fiber content reaches 0.75% (A4), the increased content causes the fibers to tangle more easily, and the appearance of voids in the fiber clusters leads to the increase of macropores. In addition, since fiber and cement particles belong to two kinds of substances, the interface cannot be completely contacted. The tiny pores between the interface lead to the increase of micropores. Therefore, the curve of A4 is generally higher than that of A2 and A3.

The *T*_2_ spectrum curve in Figure 4 more intuitively shows the freeze-thaw response characteristics of the internal structure and the influence of fiber on the frost resistance of cement mortar. In Figure 4a, with the increase of freeze-thaw cycles, the peak value of the micropore part roughly presents an increasing trend, indicating that the number of micropores keeps increasing. However, the number of macropores decreases first and then increases, which is attributed to cement hydration. Before 25 cycles, the cement hydration is still in progress, and the generated hydration products fill or separate a part of macropores into micropores, resulting in the decrease of macropores and the increase of micropores. After 25 cycles, the hydration is almost completed and the freeze-thaw effect is continuously enhanced, so the number of macropores increases significantly. In comparison, the change amplitude of the *T*_2_ spectrum curve with freeze-thaw cycles is A3 < A2 < A4 < A1, which once again confirms that the freeze-thaw resistance of cement mortar increases first and then decreases with the fiber content. The optimal content is approximately 0.5%.

### 3.2. UCS of Cement Mortar under Freeze-Thaw Cycles

Figure 6 shows the changes of UCS of cement mortars with different fiber content under freeze-thaw cycles. It can be found that the UCS curve shows an ‘upward convex’ shape with the increase of fiber content. UCS increases continuously before the fiber content reaches 0.5% and begins to decrease when the fiber content exceeds 0.5%. After analysis, it is concluded that the existence of fibers is equivalent to adding a ‘skeleton’ in the cement mortar. The cement particles are closely connected with the surface of fibers, so the fibers are pulled under the action of compressive stress, which can effectively improve the compressive strength of cement mortar. When the fiber content is high, the fiber agglomeration effect causes the fiber cluster to become a weak element existing in the cement mortar, which in turn leads to a decrease in the strength of the cement mortar. With the increase of freeze-thaw cycles, the UCS curves of four groups also show an ‘upward convex’ shape, which is the result of the interaction between cement hydration and freeze-thaw. In the early stage of freeze-thaw test, the freeze-thaw deterioration is weak and the cement hydration is strong. The resulting hydration products can improve the compressive strength of cement mortar. With the increase of freeze-thaw cycles, the freeze-thaw effect is enhanced and the hydration reaction is almost completed, so the freeze-thaw degradation effect makes the compressive strength of cement mortar continuously decrease. It is worth noting that the evolution curves of UCS under freeze-thaw conditions are different with fiber content. When the fiber content is lower than 0.25%, the turning point occurs in the 25th cycle, while when the fiber content is higher than 0.5%, the turning point occurs in the 50th cycle. According to the study of Choi et al. [33], the substance produced by the chemical reaction between clinker and fiber covering the surface of clinker will hinder the hydration reaction. Therefore, the hydration reaction time of cement will be prolonged with the increase of fiber content. When the fiber content is higher than 0.5%, the hydration reaction may not be completely stopped until 50 freeze-thaw cycles.

## 4. Discussion

### 4.1. Relationship between UCS and Porosity

Macroscopic mechanical properties are often determined by microstructure. For this reason, many scholars have tried to establish the connection between microstructure parameters and macroscopic mechanical properties, and tried to predict the strength evolution law by analyzing the change of microstructure parameters [34]. Gao et al. [35] established a strength degradation model of sandstone under freeze-thaw cycles based on porosity changes:(5)$$F(N)/F_{0}=\beta \cdot \exp\left[-\lambda (\Delta P)\right]$$
where F(N) and $F_{0}$ are strengths after 0 cycle and N cycles, respectively, β is the correction coefficient, λ is the freeze-thaw strength deterioration factor, and ΔP is the change in porosity.

Figure 7a shows the relationship between UCS and porosity of group A1 under freeze-thaw. It can be found that with the increase of freeze-thaw cycles, porosity increases continuously, while UCS increases first and then decreases, that is, there is no clear correlation between porosity and UCS in the early stage of freeze-thaw test. In fact, as shown in Figure 7b, Liu et al. [36] also proved that there was no exact relationship between porosity and UCS in their study. In Figure 7c, the degradation model proposed by Gao et al. [35] was used to fit the data in Figure 7a, and the correlation coefficient reached only 0.626. Since this model considers only the effect of porosity, it cannot reflect the increase of UCS in the initial stage. It is considered that UCS may be affected by both porosity and pore distribution. Unlike rock materials, hydration reaction is still going on in the early freeze-thaw period, and hydration products are constantly changing the pore distribution inside cement mortar. Therefore, UCS still increases despite the increase of sample porosity. From this point of view, the deterioration model established only considering the change of porosity cannot reflect the influence of pore distribution state, so there will be a great error in predicting the strength evolution law of cement-based materials.

### 4.2. NMR Fractal Characteristics

Since the pore distribution is complex and irregular, it is difficult to quantitatively characterize it using traditional Euclidean geometry. Fractal theory holds that although pore distribution is a chaotic system, it can be characterized by a stationary non-integer dimension between Euclidean dimensions, which is defined as fractal dimension [37]. Fractal dimension is a quantitative parameter that describes the degree of irregularity of fractal objects, so the complexity and irregularity of the pore structure can be indirectly reflected by this parameter. The larger the fractal dimension, the more complex the pore structure and the stronger the non-uniformity of pore distribution [38].

According to the fractal theory [39], the number of pores n(>r) with a diameter larger than r satisfies the power function relationship:(6)$$n(>r)=\int_{r}^{r_{\max}}I(r)dr=ar^{-D}$$
where $r_{\max}$ is the maximum pore size, I(r) is the pore size distribution density, a is a constant, and D is the pore fractal dimension.

The volume of pores with size less than r is expressed as:(7)$$V(<r)=\int_{r_{\min}}^{r}I(r)ar^{3}dr$$
where $r_{\min}$ is the minimum pore size.

Combining Equations (6) and (7), V(<r) can be expressed as:
(8)$$V(<r)=\beta (r^{3-D}-r_{\min}^{{}^{3-D}})$$
where β is a constant.

The cumulative pore volume fraction with pore size smaller than r is expressed as:(9)$$S_{V}=\frac{V(<r)}{V_{S}}=\frac{r^{3-D}-r_{{}^{\min}}^{3-D}}{r_{{}^{\max}}^{3-D}-r^{3-D}}$$
where $V_{S}$ is the total pore volume.

Due to $r_{\min}\ll r_{\max}$, Equation (9) can be alternatively written as:(10)$$S_{V}=\frac{r^{3-D}}{r_{{}^{\max}}^{3-D}}$$

According to Equations (2) and (10), $S_{V}$ can be expressed as:(11)$$S_{V}={\left(\frac{T_{2}}{T_{2,\max}}\right)}^{3-D}$$
where $T_{2,\max}$ is the maximum transverse relaxation time.

Take the logarithm of both sides of Equation (11):(12)$$\mathrm{lg}S_{V}=(3-D)\mathrm{lg}T_{2}+(D-3)\mathrm{lg}T_{2,\max}$$

Therefore, the fractal dimension of pore distribution can be obtained by taking the logarithm of the pore cumulative distribution curve obtained by NMR technology, and then performing linear regression on the obtained data. The slope of the regression line is 3−D.

Figure 8 shows the $\mathrm{lg}S_{V}-\mathrm{lg}T_{2}$ curve after regression. The results show that the curve is not a straight line. There are obvious differences between the micropores and macropores parts, so it is more reasonable to conduct linear regression for the two parts respectively. The fractal dimensions of micropores and macropores are denoted by $D_{\min}$ and $D_{\max}$, respectively, and the results are shown in Table 3. $D_{\max}>D_{\min}$ indicates that the complexity of macropores is higher than that of micropores. Combined with *T*_2_ spectrum curve, it can be found that the number of micropores is much larger than that of macropores. Too small of a number leads to extremely uneven distribution of macropores, so its distribution law is more difficult to characterize.

Figure 9 plots the changes of $D_{\min}$ and $D_{\max}$ with freeze-thaw cycles, respectively. As the number of freeze-thaw cycles increases, $D_{\min}$ decreases first and then increases, which is contrary to the trend of UCS. A quadratic function can be used to approximately represent the relationship between $D_{\min}$ and the number of freeze-thaw cycles N. Before 25 freeze-thaw cycles, hydration constantly changes the pore distribution state inside cement mortar, and the generated hydration products will fill some macropores into micropores, resulting in more uniform distribution of micropores and reduced complexity, so $D_{\min}$ decreases. Then, with the increase of freeze-thaw cycles, some micropores expand under the action of frost heaving force. The complexity of micropores increases instead, so $D_{\min}$ increases. In comparison, with the increase of freeze-thaw cycles, $D_{\max}$ does not show obvious regularity, except for a significant decrease at the 100th cycle. Combining with *T*_2_ spectrum distribution curve, it can be found that after 100 freeze-thaw cycles, the curve of macropore part rises significantly, that is, the number of macropores increases, and its distribution law is easier to characterize, so $D_{\max}$ decreases. In general, the evolution law of $D_{\min}$ is more relevant to that of UCS, so the deterioration model considering the distribution state of micropores will better reflect the freeze-thaw strength evolution law of cement mortar.

## 5. Freeze-Thaw Strength Degradation Prediction Model

### 5.1. Proposal of the Model

Jia et al. [40] believe that the most direct effect of freeze-thaw on porous materials is the change of porosity, and the freeze-thaw factor characterized by porosity can describe the freeze-thaw damage process of materials, namely:(13)$$W_{t}=\frac{1-P(N)}{1-P_{0}}$$
where $W_{t}$ is the freeze-thaw factor characterized by porosity. According to the above analysis, the freeze-thaw factor represented only by porosity cannot reflect the influence of pore distribution. When describing the damage evolution law of cement mortar with hydration reaction, the coefficient α needs to be introduced and the freeze-thaw factor W can be expressed as follows:(14)$$W=\alpha \cdot W_{t}$$
where α is the coefficient related to pore distribution, which is used to characterize the influence of pore distribution on freeze-thaw factor:(15)$$\alpha =1-\frac{D_{0,\min}}{D_{N,\min}}=\frac{D_{N,\min}-D_{0,\min}}{D_{N,\min}}$$
where $D_{0,\min}$ is the fractal dimension of micropores in the initial state and $D_{N,\min}$ is the fractal dimension of micropores after N freeze-thaw cycles.

According to Equations (13)–(15), the freeze-thaw factor is expressed as follows:(16)$$W=\frac{1-P(N)}{1-P_{0}}\cdot \frac{D_{N,\min}-D_{0,\min}}{D_{N,\min}}$$

Table 4 lists the changes of freeze-thaw factor W and UCS with freeze-thaw cycles. With the increase of freeze-thaw cycles, the freeze-thaw factor decreases first and then increases, while the UCS increases first and then decreases. There is a roughly negative correlation between them. Therefore, the freeze-thaw factor considering both porosity and pore distribution can effectively reflect the freeze-thaw strength deterioration of cement mortar. Figure 10 plots the change of relative compressive strength with the change amount of freeze-thaw factor. When the exponential model proposed by Gao et al. [35] is used for fitting, the fitting degree reaches 0.958. Therefore, the exponential model can also be used to characterize the relationship between the relative UCS and the change amount of freeze-thaw factor, namely:(17)$$\frac{F(N)}{F_{0}}=\eta \cdot \exp(\delta \cdot \Delta W)$$
where, η is the correction coefficient, δ is the reduction factor, and ΔW is the change amount of freeze-thaw factor, $\Delta W=W(N)-W_{0}$.

According to Equations (16) and (17), the freeze-thaw strength deterioration prediction model can be obtained as follows:(18)$$\frac{F(N)}{F_{0}}=\eta \cdot \exp[\delta \cdot \frac{1-P(N)}{1-P_{0}}\cdot \frac{D_{N,\min}-D_{0,\min}}{D_{N,\min}}]$$

### 5.2. Validation of the Model

To verify the accuracy of the prediction model proposed in this study, it is compared with models proposed by Gao et al. [35] and Deng et al. [29]. These models are shown in Table 5. The models of this study and Gao et al. [35] take ΔW and ΔP as independent variables, respectively. In order to compare the prediction effect more intuitively, ΔW and ΔP are represented by the number of freeze-thaw cycles N by regression analysis, as shown in Figure 11. ΔW and N can be fitted by quadratic polynomial, and ΔP N can be fitted by power function, namely:(19)$$\Delta W=0.002\times N^{2}-0.171\times N-0.733$$
(20)$$\Delta P=7.388\times 10^{-4}\times N^{1.878}$$

Substitute Equation (19) into Equation (17) and Equation (20) into Equation (5), and use the three prediction models to fit the data of Group A1, respectively. As shown in Figure 12, compared with the model of Gao et al. [35], the prediction results of this study and those of Deng et al. [29] can accurately reflect the strength evolution law of cement mortar under freeze-thaw cycles.

In order to improve the reliability, the data of Hu et al. [17] is fitted by using the models of this study and Gao et al. [35]. Table 6 is the research data of Hu et al. [17], and Figure 13 is the fitting result. It can be found that the prediction results of the model proposed in this study are relatively accurate, while the model of Gao et al. [35] has great discreteness because it considers only the change of porosity.

## 6. Conclusions

Taking cement mortar as the research object, this study used NMR technology to study the effect of polyester fiber on the frost resistance of cement mortar, and put forward a freeze-thaw strength prediction model based on porosity and pore distribution characteristics. The main conclusions are as follows:
(1)Fiber dispersed in cement mortar can fill the gaps between particles, so the addition of fiber can effectively improve the frost resistance of cement mortar. When the fiber content exceeds 0.5%, the fibers are easily entangled into clusters. The macropores inside the fiber clusters and the micropores at the interface between fiber and cement increase, resulting in the weakening of freeze-thaw resistance of cement mortar;(2)Cement hydration causes the UCS evolution curve of cement mortar to present the ‘upward convex’ shape under freeze-thaw conditions. Hydration reaction leads to the increase of UCS, while freeze-thaw leads to the decrease of UCS. The substances produced by the chemical reaction between fibers and clinker will prolong the hydration reaction time, resulting in UCS of samples with fiber content less than 0.5% starting to decrease after 25 freeze-thaw cycles, while that of samples with fiber content more than 0.5% starting to decrease after 50 freeze-thaw cycles;(3)Based on fractal theory, it is found that the fractal dimension of micropores *D*_min_ has a negative correlation with UCS under freeze-thaw conditions. The freeze-thaw strength prediction model considering both porosity and pore distribution can accurately reflect the strength evolution law of cement mortar under freeze-thaw cycles.

In this study, pore information was obtained and characterized by NMR technology and fractal theory, and the prediction model established based on the microstructure improved the prediction accuracy of cement mortar material strength under freeze-thaw cycles. In recent years, three-dimensional reconstruction of pore structure has become a new research direction, and using this technology to establish visual pore structure will be helpful to analyze the pore evolution process. In addition, as the most basic building materials, cement-based materials are used in various stress environments, and it is equally important to establish prediction models such as shear and tension under freeze-thaw conditions.

## Figures and Tables

**Figure 1 materials-15-07316-f001:**
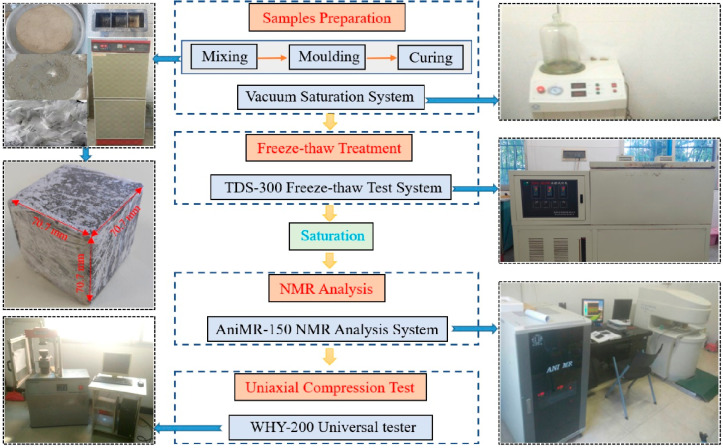
Experimental procedures and instruments.

**Figure 2 materials-15-07316-f002:**
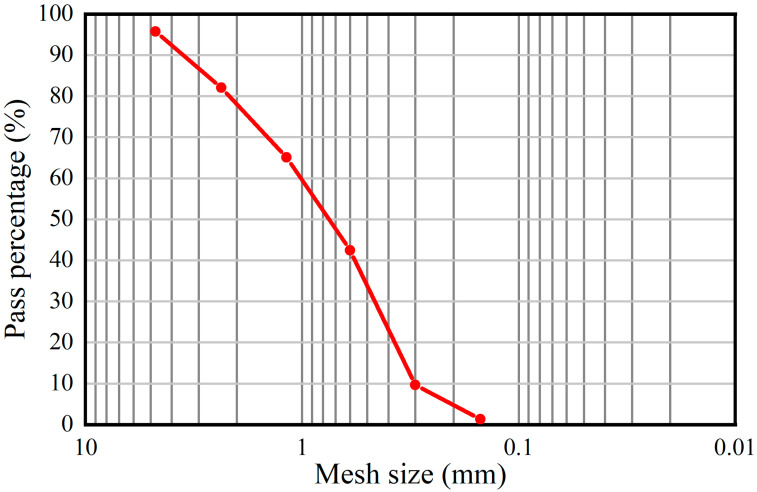
River sand particle gradation.

**Figure 3 materials-15-07316-f003:**
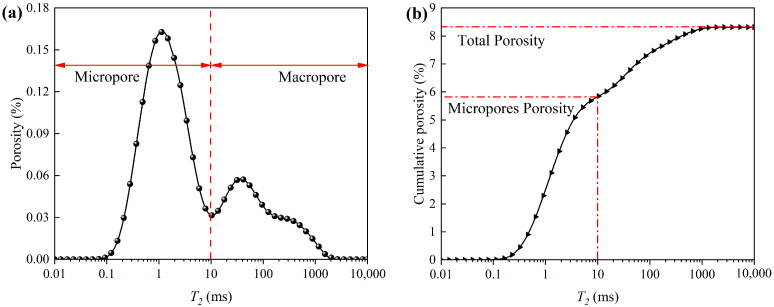
*T*_2_ spectrum and porosity cumulative distribution curve. (**a**) *T*_2_ spectrum curve, (**b**) Porosity cumulative distribution curve.

**Figure 4 materials-15-07316-f004:**
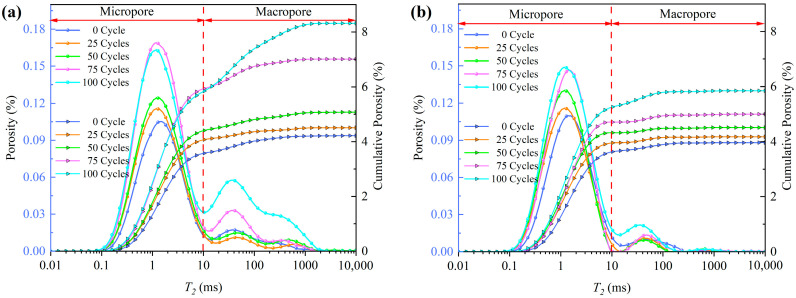
Changes of pore distribution with freeze-thaw cycles. (**a**) A1, (**b**) A2, (**c**) A3, (**d**) A4.

**Figure 5 materials-15-07316-f005:**
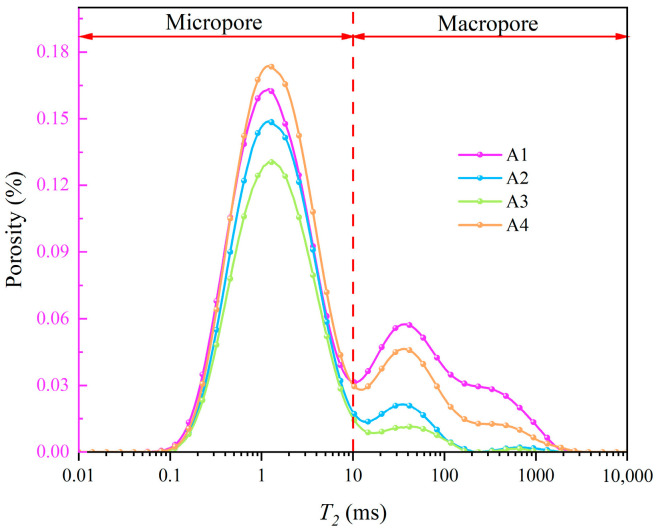
*T*_2_ spectral distribution curves of cement mortars with different fiber contents after 100 freeze-thaw cycles.

**Figure 6 materials-15-07316-f006:**
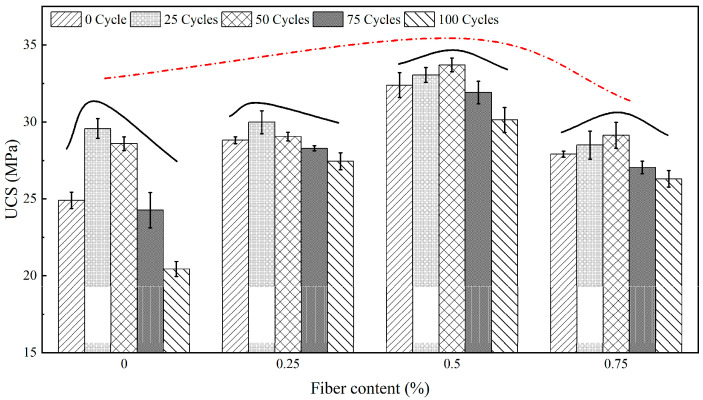
UCS of cement mortar under freeze-thaw cycles.

**Figure 7 materials-15-07316-f007:**
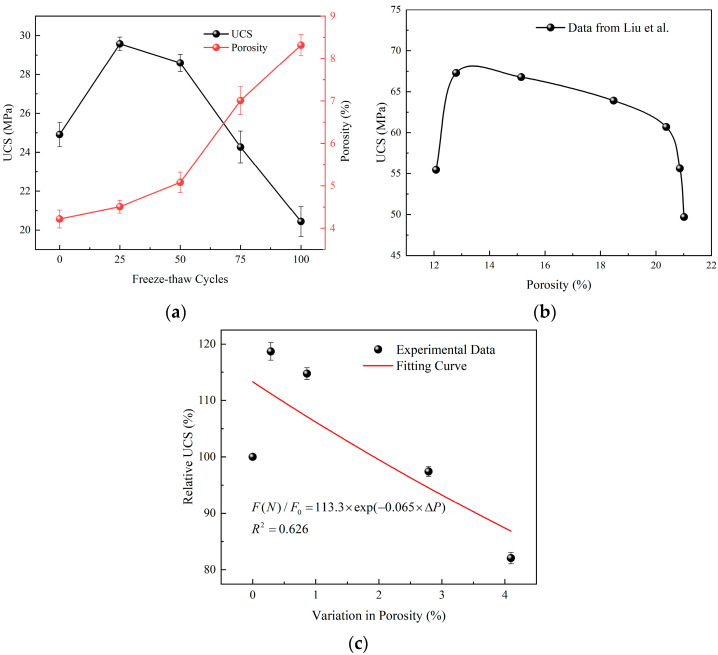
Relationship between UCS and porosity. (**a**) Variation of UCS and porosity with freeze-thaw cycles, (**b**) Research results of Liu et al. [36], (**c**) Fitting result of the deterioration model proposed by Gao et al. [35].

**Figure 8 materials-15-07316-f008:**
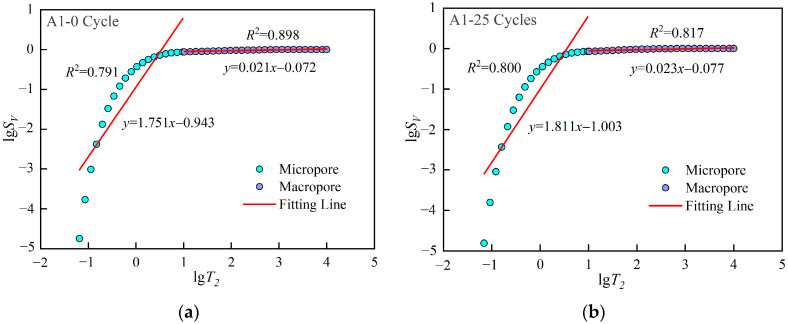
NMR fractal characteristics of samples in group A1.

**Figure 9 materials-15-07316-f009:**
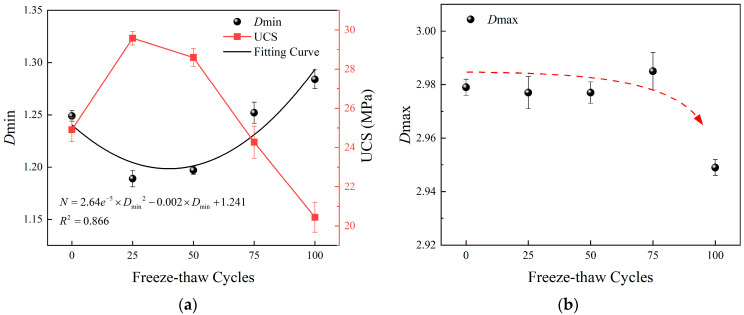
Changes of $D_{\min}$ and $D_{\max}$ under freeze-thaw cycles. (**a**) $D_{\min}$ (**b**) $D_{\max}$.

**Figure 10 materials-15-07316-f010:**
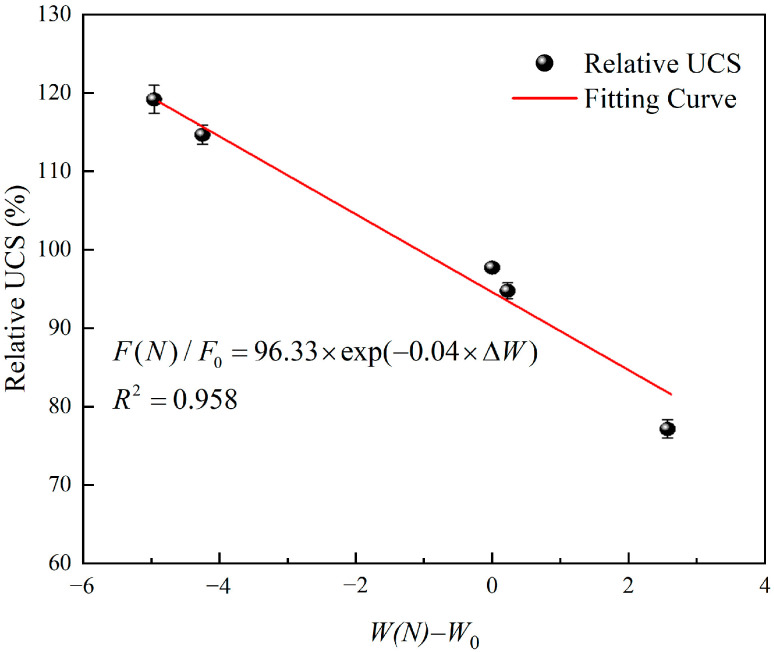
The relationship between relative UCS and the change amount of the freeze-thaw factor.

**Figure 11 materials-15-07316-f011:**
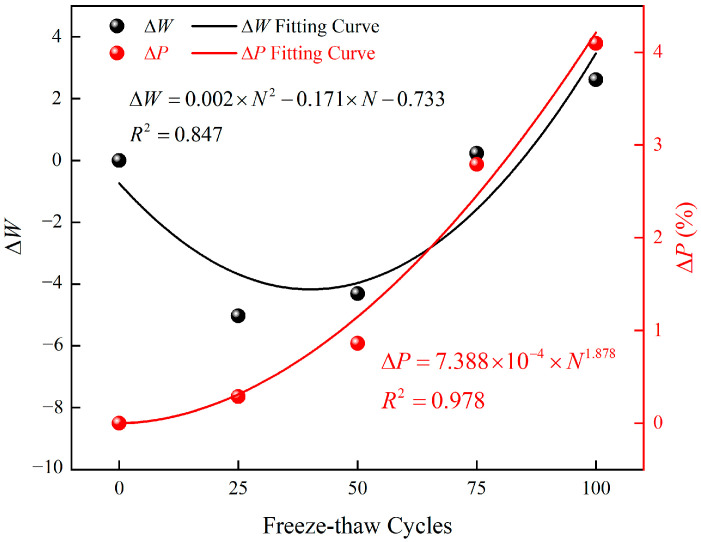
Variation of ΔW and ΔP with freeze-thaw cycles.

**Figure 12 materials-15-07316-f012:**
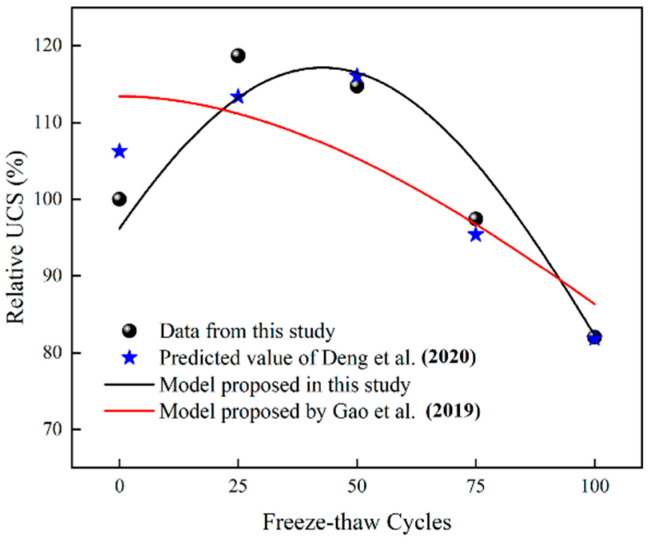
Prediction results of three models [29,35].

**Figure 13 materials-15-07316-f013:**
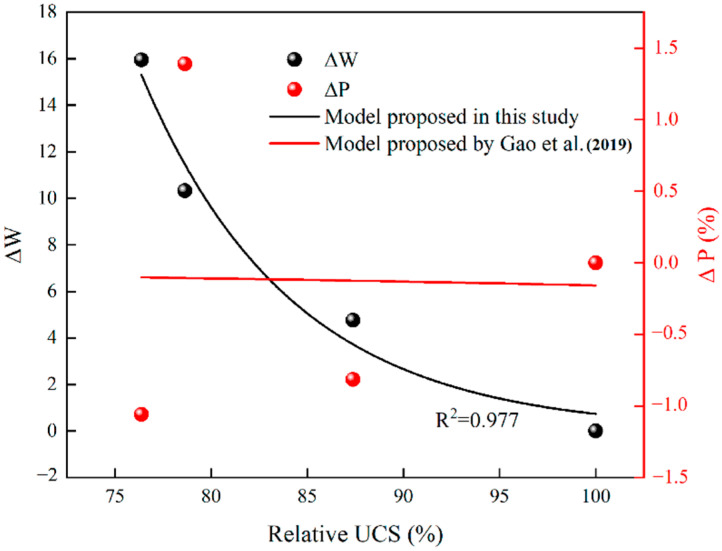
Fitting results using data from Hu et al. [17] and Gao et al. [35].

**Table 1 materials-15-07316-t001:** Material ratio table.

Group	Cement (kg/m^3^)	Sand (kg/m^3^)	Water (kg/m^3^)	Water Reducing Agent (%)	Polyester Fiber (%)
A1	360	760	162	0.5	0
A2	360	760	162	0.5	0.25
A3	360	760	162	0.5	0.5
A4	360	760	162	0.5	0.75

**Table 2 materials-15-07316-t002:** Porosity change rates of four groups after different cycles (%).

Group	Porosity Change Rate					Standard Deviation				
	0	25	50	75	100	0	25	50	75	100
A1	0	6.826	20.408	66.106	97.085	0	1.767	0.306	0.447	3.896
A2	0	5.243	13.738	25.989	47.391	0	0.587	0.954	0.435	1.132
A3	0	5.290	11.696	21.214	28.268	0	1.324	1.496	0.768	0.952
A4	0	12.111	23.658	59.083	93.528	0	0.943	1.996	2.357	1.413

**Table 3 materials-15-07316-t003:** NMR fractal dimension.

Fractal Dimension	Freeze-Thaw Cycles				
	0	25	50	75	100
*D* _min_	1.249	1.189	1.197	1.252	1.284
*D* _max_	2.979	2.977	2.977	2.985	2.949

**Table 4 materials-15-07316-t004:** Changes of W and UCS with freeze-thaw cycles.

	Freeze-Thaw Cycles				
	0	25	50	75	100
W	0	−5.03	−4.31	0.23	2.61
UCS (MPa)	24.91	29.57	28.59	24.27	20.44

**Table 5 materials-15-07316-t005:** Summary of prediction models.

Model	Formula	Note
This study	$\frac{F(N)}{F_{0}}=\eta \cdot \exp[\delta \cdot \frac{1-P(N)}{1-P_{0}}\cdot \frac{D_{N,\min}-D_{0,\min}}{D_{N,\min}}]$	Related to total porosity P and micropore fractal dimension $D_{\min}$
Gao et al. [35]	$F(N)/F_{0}=\beta \cdot \exp\left[-\lambda (\Delta P)\right]$	Related to the change in porosity ΔP
Deng et al. [29]	$F=\beta_{0}+\beta_{1}D_{\max}+\beta_{2}P_{\max}+\beta_{3}D_{\max}\times P_{\max}$	Related to macropore porosity $P_{\max}$ and macropore fractal dimension $D_{\max}$

**Table 6 materials-15-07316-t006:** The research data of Hu et al. [17].

Sample	UCS(MPa)	Porosity (%)	*D* _min_
A	1.427	15.245	1.637
B	1.247	14.431	1.718
C	1.122	16.634	1.829
D	1.090	14.187	1.943

## Data Availability

Data sharing is not applicable in this article.

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
