# Peer review of "The Freeze-Thaw Strength Evolution of Fiber-Reinforced Cement Mortar Based on NMR and Fractal Theory: Considering Porosity and Pore Distribution"

_materials, 2022, doi:10.3390/ma15207316_

Round 1

Reviewer 1 Report

    The main objective of this paper is to predict the free-thaw strength/resistance of the cement mortar by using nuclear magnetic resonance technology, analyze the changes that occurred on a microscopic pore scale, and investigate the effect of polyester fiber on the frost resistance of cement mortar.Thus, the article content is actual and adequate to the research article.

    Abstract is adequate to article content

    Keywords are correctly proposed.

    The literature review is based on 17 references, and they are related to the article content; however,  it needs to add more references.

 However, I indicate some issues that require additional clarification:

#1 Extend introduction

Please extend the introduction with a summary of the article organization. I mean “section 2 concerns…, section 3 presents…, etc.” It is noted that previous studies reviewed are very few, and it is recommended to increase the previous studies to clarify the gap.

#2 Results

The results in the article are well presented. This is a strong element of the article, but the figures presented need to be aligned with the journal style ; please see ( https://www.mdpi.com/journal/materials/instructions ).

#3 Discussion

The results were discussed well, but it needs to be compared with previous studies, if any. Furthermore, section 4.2. titled with  NMR Fractal Characteristics need to extend the discussions, especially since many results were obtained related to it.

#4 language

There are several grammatical and typo errors present that impede the flow of reading. The English language needs to be improved for the whole manuscript.

#5 Conclusion

Please extend the conclusions section with future research direction

Reviewer 2 Report

This paper needs to be revised. However, some comments need to be addressed:

·         The objective and study motivation need to be shown in the introduction section. 

·         The authors pointed out the results in Figure 7 in Section 4.1, which seems to be confusing, and then show other results based on Gao et al.2019 (23)? Could you please elaborate on this? 

·         The authors should compare this study with previous literature as a benchmark and put it either in a chart or a table. 

·         The conclusion of this paper was very short. Extend this part by showing the results and recommending any suggestions for future work, if any.

Reviewer 3 Report

This work presents an extensive work on the behavior of fiber-reinforced cement mortar and proposes a model based on fractal theory for improving the description of some important material parameters. 

Some issues should be addressed prior to publication:

 1. Improve resolution of Figure 1. Both photographs and block diagram seem pixelated. In general quality of the photographs should be improved.

 2. Correct the format of Table 1. The first ammount of water is misplaced, there is an extra line between A1 and A2 data.

3. Lines 125-126. Seems like one or two words are missing in the text. Please check and correct

4.  Figure 4. When comparing plots, could be better to use the same scale on both axes. Vertical axis is changing in each plot.

5. lines 165-167. Check the writing of the second sentence. 

6. Table 2. Values of Amust include an uncertainty in order to be compared properly. At least discuss in the text the uncertainty calculation for these values.

7. Lines 176-179. Where do you explain how the "porosity change rate" is calculated?

8. Figure 4 + Figure 5. References to figure 4 and 5 seems to go back and forth. It is unclear the construction of figure 5. Seems like data displayed is already in figure 4. 

Figure 7.  Figure 9. Figure 10.  Consider adding uncertainties to the experimental data

Reviewer 4 Report

In this manuscript, the authors systematically analyzed the influence of porosity and pore size distribution on mechanical properties of fiber-reinforced cement. The authors firstly quantified the changes of porous structure in freeze-thaw test and determined the optimal fiber content. Then the authors analyzed uniaxial compressive strength (UCS) under different freeze-thaw cycles. A correlation between UCS and fractal dimensions of micropores was found, and prediction model on UCS was proposed. Overall, this manuscript is interesting, and it enhances understanding the influence of nanostructure on cement’s mechanical properties. My only concern is whether their analysis can be generalized. Some clarification maybe needed. I recommend a major revision on this manuscript.

(1)    The unit to T2 is missing in Figure 3-5.  

(2)    The authors should specify the condition of samples showed in Figure 5.

(3)    Although the authors show better fitting result than Gao et al.’s model, has they validated their model on Gao or others’ data? The number of data points is very limited in this research, and model may overfitted to the authors’ data.

Round 2

Reviewer 1 Report

I appreciate the efforts carried out by the authors that led to an excellent improvement of the manuscript. Hence, I recommend accepting the manuscript for publication in its current form. 

Best regards

Author Response

Your comments are very helpful to our article revision. Thank you very much.

Yours sincerely

Reviewer 2 Report

I need clarification from the authors. Why do they have so many citations from their own references?

Author Response

Thank you for your suggestion. Since this study focuses on freeze-thaw and NMR technology, our previous articles described the damage mechanism of freeze-thaw and the principle of NMR technology, so we cited them here. Of course, in order to improve the objectivity of the article, we have appropriately deleted our own references and cited the research results of others as much as possible.

‘Liu, T.; Zhang, C.; Zhou, K.; Tian, Y. Freeze-Thaw Cycling Damage Evolution of Additive Cement Mortar. Eur. J. Environ. Civ. Eng. 2021, 25, 2089-2110.

Liu, T.; Zhang, C.; Li, J.; Zhou, K.; Ping, C. Detecting freeze–thaw Damage Degradation of Sandstone with Initial Damage using NMR Technology. B. Eng. Geol. Environ. 2021, 80, 4529-4545.’

The contents cited in the above two references can be reflected by the research results of others, so they are deleted.

Reviewer 3 Report

Most of the issues were addressed in a proper way. Conclusions are still somehow weak, but better than in the previous version. Results are better presented now.

Author Response

Thanks for your suggestion, we have further revised the conclusions as follows:

‘Taking cement mortar as the research object, this study used NMR technology to study the effect of polyester fiber on the frost resistance of cement mortar, and put forward a freeze-thaw strength prediction model based on porosity and pore distribution characteristics. The main conclusions are as follows:

(1) Fiber dispersed in cement mortar can fill the gaps between particles, so the addition of fiber can effectively improve the frost resistance of cement mortar. When the fiber content exceeds 0.5%, the fibers are easily entangled into clusters. The macropores inside the fiber clusters and the micropores at the interface between fiber and cement increase, resulting in the weakening of freeze-thaw resistance of cement mortar.

(2) Cement hydration causes the UCS evolution curve of cement mortar to present the ‘upward convex’ shape under freeze-thaw conditions. Hydration reaction leads to the increase of UCS, while freeze-thaw leads to the decrease of UCS. The substances produced by the chemical reaction between fibers and clinker will prolong the hydration reaction time, resulting in UCS of samples with fiber content less than 0.5% starting to decrease after 25 freeze-thaw cycles, while that of samples with fiber content more than 0.5% starting to decrease after 50 freeze-thaw cycles.

(3) Based on fractal theory, it is found that the fractal dimension of micropores Dmin has a negative correlation with UCS under freeze-thaw conditions. The freeze-thaw strength prediction model considering both porosity and pore distribution can accurately reflect the strength evolution law of cement mortar under freeze-thaw cycles.

In this study, pore information was obtained and characterized by NMR technology and fractal theory, and the prediction model established based on the microstructure improved the prediction accuracy of cement mortar material strength under freeze-thaw cycles. In recent years, three-dimensional reconstruction of pore structure has become a new research direction, and using this technology to establish visual pore structure will be helpful to analyze the pore evolution process. In addition, as the most basic building materials, cement-based materials are used in various stress environments, and it is equally important to establish prediction models such as shear and tension under freeze-thaw conditions.’